# MetaSpot: A General Approach for Recognizing the Reactive Atoms Undergoing Metabolic Reactions Based on the MetaQSAR Database

**DOI:** 10.3390/ijms241311064

**Published:** 2023-07-04

**Authors:** Angelica Mazzolari, Pietro Perazzoni, Emanuela Sabato, Filippo Lunghini, Andrea R. Beccari, Giulio Vistoli, Alessandro Pedretti

**Affiliations:** 1Dipartimento di Scienze Farmaceutiche, Università degli Studi di Milano, Via Luigi Mangiagalli, 25, I-20133 Milano, Italy; angelica.mazzolari@unimi.it (A.M.); pietroperaz@libero.it (P.P.); emanuela.sabato@unimi.it (E.S.); giulio.vistoli@unimi.it (G.V.); 2EXSCALATE, Dompé Farmaceutici S.p.A., Via Tommaso De Amicis, 95, I-80131 Napoli, Italy; filippo.lunghini@dompe.com (F.L.); andrea.beccari@dompe.com (A.R.B.)

**Keywords:** metabolism prediction, site of metabolism, MetaQSAR, random forest, atom typing

## Abstract

The prediction of drug metabolism is attracting great interest for the possibility of discarding molecules with unfavorable ADME/Tox profile at the early stage of the drug discovery process. In this context, artificial intelligence methods can generate highly performing predictive models if they are trained by accurate metabolic data. MetaQSAR-based datasets were collected to predict the sites of metabolism for most metabolic reactions. The models were based on a set of structural, physicochemical, and stereo-electronic descriptors and were generated by the random forest algorithm. For each considered biotransformation, two types of models were developed: the first type involved all non-reactive atoms and included atom types among the descriptors, while the second type involved only non-reactive centers having the same atom type(s) of the reactive atoms. All the models of the first type revealed very high performances; the models of the second type show on average worst performances while being almost always able to recognize the reactive centers; only conjugations with glucuronic acid are unsatisfactorily predicted by the models of the second type. Feature evaluation confirms the major role of lipophilicity, self-polarizability, and H-bonding for almost all considered reactions. The obtained results emphasize the possibility of recognizing the sites of metabolism by classification models trained on MetaQSAR database. The two types of models can be synergistically combined since the first models identify which atoms can undergo a given metabolic reactions, while the second models detect the truly reactive centers. The generated models are available as scripts for the VEGA program.

## 1. Introduction

The in silico evaluation of the ADME/Tox (absorption, distribution, metabolism, excretion, and toxicity) profile for new chemical entities has attracted great interest in the last decades since this represents a rapid and cost-effective strategy to discard molecules with unfavorable drug-like properties at the early stages of the drug discovery process [1,2]. These preliminary filtering approaches have the clear objective to reduce attritions and risk of failures. In this way, the attrition rate related to pharmacokinetic issues decreased from about 40% in the 1990s to less than 10% during the 2000s [3].

The prediction of drug metabolism has played a crucial role as the involved metabolic reactions and the formed metabolites influence the overall pharmacokinetic profile of any xenobiotic compound determining, above all, its efficacy and safety [4]. In the past, the most common computational approaches for drug metabolism were focused on the prediction of the metabolites formed by specific enzymes and were mostly based on tailored docking simulations (the so-called local methods) [5]. The redox reactions catalyzed by cytochromes P450 greatly enjoyed these studies [6,7] and some effective approaches were implemented into commercial products [8]. In contrast, global methods able to predict the entire metabolic fate of a given molecule were less frequent and mostly based on sets of knowledge-based rules [9]. The scarcity of global methods might be attributed to the lack of extended metabolic datasets amenable for such predictive studies [10].

The situation evolved in the last years due to the general increase of the available scientific data which also concerned the field of drug metabolism. The recent progress in artificial intelligence methods also played a crucial role offering convenient approaches for the metabolism prediction [11]. Among the recent in silico global methods published in this framework, one may mention (a) Biotransformer, which combines knowledge-based rules with machine learning approaches to predict phase I and phase II metabolic reactions as well as human gut and environmental microbial reactions [12]; (b) FAME, which predicts the sites of metabolism (SOM) for phase I and phase II reactions based on tree classifiers [13], and (c) Xenosite, which predicts the outcomes of some relevant metabolic reactions by using neural network techniques mainly based on quantum-mechanical descriptors [14].

Notably, the development of reliable predictive models requires highly accurate metabolic data because even some inaccurate data can undermine the overall predictive power of the resulting models. Unfortunately, several available metabolic resources do not possess a suitable accuracy since they are collected by automatic interrogation of other databases [15] and often combine xenobiotic and endogenous reactions with a metabolomics perspective [16]. Thus, we recently proposed a manually annotated database, MetaQSAR, collected by critical meta-analyses of recent literature [17]. The MetaQSAR database implements an ad hoc classification which subdivides the metabolic reactions into 3 major classes, 21 classes and 101 subclasses with a hierarchic structure which allows a precise characterization of each reported biotransformation [18].

MetaQSAR proved successful in developing predictive models for two conjugation reactions (glucuronidation and reaction with glutathione), which play crucial roles in determining the safety of chemicals [19,20]. Datasets extracted from MetaQSAR were utilized to develop the above-mentioned FAME method [12], thus suggesting a general applicability of its metabolic data. The global applicability was further confirmed by a recent study, in which MetaQSAR-based datasets allowed the development of models to predict the occurrence of almost all the reaction classes plus some suitably populated subclasses by using the Random Forest (RF) classification algorithm. These predictive models were implemented in a freely released tool, MetaClass, able to predict which metabolic reactions undergo an input molecule [21].

The here-reported study can be seen as a step further since it is based on the same MetaQSAR-based datasets, and its aim involves the development of RF-based models to recognize the specific sites of metabolism (SOM) for all the reaction classes and subclasses already considered by the MetaClass approach. Similarly to what was already done for MetaClass, the here-developed predictive models are released into a suite of freely available scripts, called MetaSpot, to predict which atoms undergo specific metabolic reactions for a given input substrate.

## 2. Results

### 2.1. MetaQSAR-Based Datasets

The study was based on the same first-generation metabolic reactions utilized by the previous study [20]. Specifically, the analyses involved 3788 metabolic reactions, which are classified into 3 major classes (i.e., redox reactions, hydrolyses and conjugations), 21 classes and 101 subclasses. Predictive models were generated for all classes and subclasses which include at least 50 metabolic reactions. For each simulated dataset, attention was focused only on the molecules which undergo the corresponding metabolic reaction and considering as reactive atoms (R), the atoms involved in the biotransformation and as non-reactive atoms (NR) all the remaining atoms without exceptions. The choice of focusing on the substrates to collect the non-reactive atoms had two major reasons: (a) avoiding extremely unbalanced datasets (and however the number of NR atoms is still markedly higher); (b) minimizing the number of false negative NR atoms.

For each class and subclass, the performed predictive analyses can be subdivided into two steps. In the first step, the utilized datasets comprised all collected NR and R atoms. Each dataset was balanced by randomly undersampling the majority NR class and this task was repeated 100 times to minimize the randomness. The second step was based on specifically collected datasets comprising only atoms of the same atom type(s) (according to Kier-Hall E-states [22]) of the reactive centers to assess the capacity of the developed models to discriminate between reactive and the non-reactive centers of the same atom type. The models developed by the two steps can be sequentially utilized since the models of the first step are intended to perform a coarse filter to discard those atoms which cannot undergo a given biotransformation, while the models of the second round are more finely tuned to recognize the truly reactive centers.

### 2.2. Prediction of the Reactive Atoms for the Reaction Classes

Figure 1 and Appendix A report the performances for the considered classes of metabolic reactions as reached by the first round of predictions in which the NR atoms are randomly extracted from all the possible NR atoms belonging to the collected substrates. Appendix A reports the overall performance as computed by considering both classes. For each metabolic reaction, Appendix A comprises the mean and range values of the performance metrics as obtained by repeating the model generation 100 times. For the three major classes of reactions, Figure 1 and Appendix A also include the corresponding mean values as well as the overall means.

Figure 1 reveals satisfactory performances for almost all considered metabolic reactions with the conjugations which afford, on average, the best results, while redox reactions and hydrolyses yield slightly worse performances. The reported accuracy values indicate that, on average, 90% of reactive atoms are correctly predicted. Only two reactions classes show accuracy values comprised between 0.7 and 0.8 with 11 out of 17 accuracy values being greater than 0.9. The MCC values also show similar trends: 11 out of 17 metabolic reactions show MCC ≥ 0.8 and only 2 classes show MCC < 0.5.

Overall, the models appear to be quite stable as confirmed by the accuracy ranges (see Appendix A), which are lower than 0.02 in 9 cases out of 17, as well as by the MCC ranges, which are lower than 0.15 for 10 classes. Only two classes, i.e., redox of quinones and other hydrolyses, show modest performances in terms of both mean and range values. The obtained outcomes do not reveal significant correlations between performance and number of instances in each class. This suggests that all the considered classes are populated enough to develop reliable models. The observed performance differences are thus ascribable to the intrinsic complexity of each metabolic reaction rather than to the chemical space covered by its substrates.

Figure 2 and Appendix A report the analysis of the feature importance for the 17 predicted biotransformations: Appendix A compiles the selected descriptors for each model, while Figure 2 focuses on their frequencies. The first consideration involves the number of selected features which ranges from 1 to 11 with an average value equal to 4.8. The rather limited number of included variables should assure a reliable robustness for the generated models avoiding overfitting biases. Figure 2 emphasizes the remarkable role played by atom typing as evidenced by the Kier and Hall E-states [21], which appear in 12 models out of 17. In fact, the Broto’s and Moreau’s atomic increments for lipophilicity [23] can also be seen as a sort of atom typing in which the atom types are indirectly encoded by their atomic increments. Indeed, replacing the Broto’s and Moreau’s atomic constants with the corresponding atom types provides truly superimposable overall performances (results not shown). As a matter of fact, all models include at least one atom type and 7 models out of 17 include both atom types.

Among the stereo-electronic parameters, atomic self-polarizability (π^S^) and atomic charge are the most frequently included descriptors. The role of atomic self-polarizability agrees with previous studies which exploited such a descriptor to predict the sites of metabolism [24] and, more in general, to rationalize the chemical reactivity. Concerning the physicochemical descriptors, the number of H-bonding groups plays a key role reasonably as it encodes for the substrate’s capacity to interact with the enzymes. In eight models, there is at least one parameter related to molecular size, which may describe the capacity of a given molecule to be accommodated within the enzymatic cavity. The marginal role of the virtual log P descriptor (see Appendix A) can be explained by considering that lipophilicity is already encoded by Broto’s and Moreau’s atomic increments.

Taken together, the obtained performances suggest that satisfactory predictions can be achieved by correctly recognizing the atom types which can undergo a given metabolic reaction. Even though almost all models also include stereo-electronic and/or physicochemical descriptors which should encode for the intrinsic reactivity and accessibility of each atom, the primary role played by atom typing might result in a high number of false positives since there is the risk that all atoms belonging to a given atom type are predicted as reactive regardless of their actual reactivity. Hence, the second round of predictions was performed by excluding atom typing to assess whether the stereo-electronic and physicochemical descriptors alone can recognize the reactive atoms regardless of their type. To do this and for each considered class, a balanced dataset was generated by randomly collecting only NR atoms having the same type(s) of the reactive sites.

Figure 1 and Appendix A report the performances obtained by this second round of predictions and compare them with those reached during the first round of predictions. While the performances of the first round were similarly satisfactory with limited differences between the considered classes, the second round reveals significant differences, and some models prove to be unsatisfactory. In detail, Figure 1 and Appendix A show that: (a) the sites of metabolism of the redox reactions (especially those involving carbon atoms) can successfully be recognized, (b) the reactive atoms of hydrolyses are recognized although with slightly worse performances compared to Appendix A especially for the hydrolysis of esters, while (c) the sites of conjugations are predicted with difficulty and the developed models show markedly worse performances compared to the first round of predictions. Overall, only 3 reaction classes show enhanced performances in this second round and the drop in the MCC value is greater than 0.4 in 6 cases. The accuracy values indicate that there are 3 classes for which the correctly identified reactive atoms are lower than 70%. In detail, the reported performances evidence that the sites of glucoronidation cannot be conveniently predicted and the redox reactions on nitrogen atoms are predicted with marked difficulty.

Figure 2 and Appendix A highlight the role of the selected features for this second round of predictions.In addition, the developed models include a limited number of variables which range from 3 to 9 with an average value equal to 4.9. Concerning the stereo-electronic parameters, these models confirm the major role of self-polarizability and atomic charges, while greater differences are seen among the physicochemical descriptors. Figure 2 emphasizes the key role of descriptors related to H-bonding as confirmed by the PSA parameter which is included into 10 models. Descriptors related to the molecular size also play a remarkable role since they are included in 15 models out of 17.

Taken together, the results of this second round of predictions suggest that the reactive atoms of redox reactions largely depend on stereo-electronic factors, which are conveniently captured by the considered descriptors. In contrast, reactive atoms involved in hydrolyses and conjugations seem to be influenced by different factors not completely encoded by the considered descriptors. Future studies involving additional descriptors and/or docking simulations could enhance the predictive models for these classes.

### 2.3. Prediction of the Reactive Atoms for the Reaction Subclasses

Figure 3 and Appendix A show the performances for the considered subclasses as reached by considering all possible NR atoms in the first round of predictions. A bird’s eye view of the obtained results reveals the subclasses provide satisfactory performances in agreement with those already reported for the metabolic classes. The remarkable results are confirmed by the accuracy values which indicate that the correctly predicted centers are ≥90% for 16 subclasses out of 23, while the accuracy is less than 0.8 only in two cases. MCC values provide similar trends with 15 out of 23 MCC values greater than 0.8. Only the oxidation of aryls and phenols and the addition−elimination reactions of glutathione yield less than satisfactory models. The comparison between Figure 1 and Figure 3 shows that the finer classification by subclasses has a beneficial role for the hydrolyses, while redox reactions and conjugations show roughly comparable performances. The models for subclasses also reveal an appreciable stability as evidenced by the accuracy range which is <0.1 for 15 cases out of 19 (see Appendix A).

Appendix A and Figure 4 report the results of the feature importance for the 19 considered subclasses: Appendix A compiles the selected descriptors, and Figure 4 shows their frequencies. Here also, the number of involved descriptors is rather limited, a fact that should avoid overfitting conditions. In detail, this ranges from 1 to 9 variables with an average equal to 5.0. The analysis of the selected features further confirms the remarkable role of the atom typing, since all models include either Kier-Hall E-states or Broto’s and Moreau’s atomic increments, and 12 models out of 19 include both atom types. On average, these results emphasize a greater role of the atom typing in subclasses compared to classes. This finding can be explained by considering that the finer classification of the metabolic reactions enhances the capacity of the atom types to detect the reactive centers.

Concerning the stereo-electronic descriptors, the developed models confirm the key role of atomic charges and self-polarizability (π^S^) in describing the chemical reactivity. In addition, the HOMO/LUMO energies appear 13 times, thus underlining the relevant role played by nucleophilicity and electrophilicity in drug metabolism. The frequencies of the physicochemical descriptors agree with what was observed in the previous models and emphasize the marked role of H-bonds and molecular size (the related descriptors appear 10 times in both cases) which encode for interacting capacity and accessibility, respectively.

Figure 3 and Appendix A detail the performances reached by the subclasses in the second round of model generation in which the utilized datasets include only NR atoms of the same type(s) of the reactive centers. As already observed above, the second round of predictions reveals marked differences in the performances of the considered subclasses. On average, the reactive atoms for the redox reactions are conveniently predicted with almost all accuracy values greater than 0.8 and 4 subclasses (out of 12) show even better performances compared to the first round. Appendix A confirms that the atoms undergoing redox reactions can be suitably recognized based on their stereo-electronic features and the finer classification of these reactions (as done in subclasses) improves the recognition of the reactive atoms.

The finer classifications of hydrolyses do not improve the resulting performances, which remain for all considered subclasses around 0.75 (as already seen in Appendix A). This output suggests that the detection of the labile functional groups (as made by atom types) plays a key role in determining the performances for these metabolic reactions, and the stereo-electronic properties are not completely effective in recognizing the labile moieties. Regarding glucuronidations, Figure 3 reveals that a finer classification improves the performances for predicting metabolic reactions of phenols and carboxylic acids, while the reactions of alcohols and the N-glucuronidations are still unsuitably predicted. Finally, Figure 3 shows that the poor performance observed for the conjugations with GSH in Figure 1 is ascribable to the reactions which occur via GSH addition and elimination, while the nucleophilic GSH additions can be conveniently predicted.

As previously seen, Appendix A and Figure 4 detail the feature importance for this second round of predictions on the 19 considered subclasses. This analysis confirms the pivotal role of atomic charges and self-polarizability (π^S^) and evidences the enhanced relevance of the atomic charge density the role of which in predicting chemical reactivity is well-documented in literature [25]. Among the physicochemical descriptors, Figure 4 emphasizes the relevance of H-bonds (as especially encoded by PSA) and molecular size with 15 occurrences in both cases.

## 3. Discussion

Since both studies are based on the same MetaQSAR-based training sets and roughly involve the same set of descriptors, Figure 5 compares the performances (as expressed by the MCC values) of the here reported models to recognize the reactive atoms with those reported in the previous MetaClass study to predict the occurrence of a given metabolic reaction. For simplicity, the analysis is focused on the predictive models as generated for the classes of metabolic reactions and compares the MetaClass performances with those here reported for both rounds of predictions.

Figure 5 clearly emphasizes that the reactive atoms involved in most metabolic reactions can be conveniently recognized by RF-based classifiers in which atom typing plays a key role (first round). For almost all classes these models outperform the other predictions suggesting that atom typing has a markedly greater role in the recognition of reactive atoms rather than in predicting the occurrence of a given biotransformation. The relevance of atom typing comes as no surprise when considering that most available global methods were based on knowledge-based rules and atom typing is probably the most effective way to transform these qualitative rules into computationally tractable molecular descriptors. Even though all the proposed models also include stereo-electronic descriptors to account for the chemical reactivity of each atom, the key role played by atom types can lead to a significant number of false positive since the models tend to predict that all atoms of a given type can yield the corresponding metabolic reaction regardless of their reactivity. Stated differently, the models based on atom typing prove successful in discarding the atoms which cannot undergo a given biotransformation, while being less performing in recognizing the truly reactive atoms among those which can undergo a given metabolic reaction.

Interestingly, the second round of prediction which do not include the atom typing, provide, on average, worst performances which shows a certain degree of similarity with those reached by the MetaClass models. This confirms that atom typing has a very modest role in predicting the occurrence of a given metabolic reaction and both sets of models similarly depend on the included physicochemical and stereo-electronic descriptors. While showing lower MCC values, many developed models of the second round exhibit satisfactory performances thus indicating that the recognition of the sites of metabolism based on reactivity descriptors is yet possible and deserves further efforts to enhance the resulting performances, especially for those classes and subclasses which provided here unsatisfactory predictive models (see above).

Indeed, and while considering the generally satisfactory results reported here, this study can be seen as an encouraging starting point which can be surely enhanced by extending the number and the informative richness of the considered descriptors. The descriptors utilized in this study were indeed chosen to cover a significant space of the (structural, physicochemical and stereo-electronic) molecular properties while being computationally simple and not time-expensive. This choice was made by considering the high number of substrates included in the MetaQSAR-based datasets and to develop simple predictive models to be quickly applied to new molecules. Notwithstanding this, one may figure out that a wider and tailored set of descriptors could yield improved predictive models. For example, the observed relevance of various stereo-electronic descriptors (here computed by semi-empirical calculations) suggests that the inclusion of similar parameters computed by DFT methods, although computationally markedly more demanding, might have a beneficial effect. The role played in many models by molecular descriptors encoding for the interaction capacities (e.g., H-bonding, molecular size and accessibility) suggests that targeted docking simulations could describe the binding process in a more direct and informative way. The enrichment of the considered descriptors may involve also the atom typing by including extended atom types and/or fingerprints. Nevertheless, the enrichment of this kind of descriptors should be cautiously pursued since an increased role of atom typing could increase the number of false positives.

In more detail, the results of this study (especially concerning the second round of predictions) can be summarized as follows:(1)Redox reactions on carbon atoms are conveniently predicted even without atom typing; this outcome indicates that their reactive atoms mostly depend on the considered stereo-electronic descriptors (e.g., atomic charges and self-polarizability).(2)Redox reactions involving nitrogen atoms (and to minor extent Sulphur atoms) are satisfactorily predicted only when using atom types, thus suggesting that their reactivity depends on different factors compared to carbon atoms.(3)Hydrolysis reactions yield markedly more accurate predictions when including atom types which reasonably allow an easy detection of the labile groups.(4)Conjugations with glucuronic acid are substantially unpredictable without atom types; this finding suggests that the reactivity of the involved centers depends on stereo-electronic factors not included in the considered descriptors.(5)Most reactions with glutathione can be successfully predicted even without atom types, a result which can be explained by considering that these conjugations depend on the here parameterized electrophilicity of the reactive centers.

As mentioned above, the two sets of predictions (i.e., with and without atom types) can be synergistically combined since the models including atom types can be used to exclude the atoms which cannot undergo the considered biotransformation, while the models without atom types should recognize the truly reactive centers.

## 4. Materials and Methods

### 4.1. Utilized Metabolic Data

The study utilized the same first-generation metabolic reactions extracted from the MetaQSAR database [17] and already utilized in the previous MetaClass study [20]. In detail, they include 3788 metabolic reactions, which involve 2787 different substrates. As done in the previous study, all the collected metabolic reactions were utilized during the model generation to cover the widest possible chemical space of the substrates. Based on the MetaClass results, the substrates are here simulated in their neutral form. A detailed description of the set-up of the collected substrates and the calculation of the utilized descriptors can be found elsewhere [20]. Briefly, the compounds were optimized by the PM7 semi-empirical method which also allowed the calculation of an extended set of stereo-electronic descriptors. For each molecule, a set of physicochemical and geometrical descriptors were calculated by the VEGA suite of programs [26]. Atom typing involved the Kier-Hall E-states [21] as well as the lipophilic atomic increments of Broto and Moreau [22].

### 4.2. Generation of Models with Atom Typing (First Round)

As described above, the predictive analyses involved all the classes and subclasses, including at least 50 instances. For each case, two predictive analyses were performed including or excluding the atom types among the utilized descriptors. The analyses including atom types were performed by using an ad-hoc script for the VEGA environment that: (a) extracts from MetaQSAR the substrates for a given class or subclass; (b) identifies within the substrates the reactive and non-reactive atoms and (c) calculates for the included substrates a set of atom- and ligand-based descriptors. For each considered class and subclass, the analyses comprised all collected atoms and involved the following tasks carried out by the above-mentioned script: (a) performing the feature selection to reduce the number of considered descriptors; (e) randomly under-sampling the NR atoms to obtain balanced datasets; (f) developing the corresponding predictive model by using the RF algorithm. Considering the very high number of NR atoms in all performed analyses, tasks (e) and (f) were repeated 100 times to minimize the randomness of the obtained results. Here, and in the analyses without atom types, the models were developed by using the RF algorithm, as implemented in the WEKA 3.8.6 software [27], by applying the default parameters since, in the previous MetaClass study, these conditions afforded the best results [20]. The feature selection was also carried out by WEKA based on both the BestFirst and the WrapperSubsetEval algorithms, as previously described [20]. All the models generated in the first round of predictions were transformed in C-based scripts by using the Tree2C program [28]. Briefly, Tree2C converts a tree model as generated by WEKA into a C-based script for the VEGA program. The so-generated script predicts the reactive centers for the molecule loaded within the VEGA workspace by applying the tree model and by calculating on the fly all the required descriptors. All the so-generated models were available within the Appendix A.

### 4.3. Generation of Models without Atom Typing (Second Round)

To assess the capacity to predict the reactivity of the sites of metabolism regardless of their atom types, the second set of predictions was developed by considering the NR atoms with the same atom type(s) of the reactive centers. For each analyses class and subclass, a specific dataset was manually collected by combining the reactive atoms with an equal number of randomly selected non-reactive atoms having the same atom type(s) of the reactive ones. The so-collected dataset undergo model generation and feature selection by adopting the same procedures described above for the first round of predictions.

## 5. Conclusions

The present study describes the MetaSpot approach, which is based on the metabolic data extracted from the MetaQSAR database and allows the predictions of the reactive centers for almost all metabolic reactions. Apart from a few documented exceptions, all the models developed within the MetaSpot project provided satisfactory performances (even without atom typing), emphasizing the possibility of conveniently recognizing the reactive atoms. The MetaSpot project can be seen as the natural extension of the already published MetaClass study [20] since MetaClass predicts which metabolic reactions undergo a given molecule, while MetaSpot predicts which are the involved sites of metabolism. The two approaches can be combined and can have a mutually validating role since the reactive centers predicted by MetaSpot for a given biotransformation can be confirmed if MetaClass predicts that the substrate can undergo the biotransformation. Similarly, a metabolic reaction predicted by MetaClass can be confirmed if MetaSpot finds at least one reactive atom for this reaction.

As detailed in the discussion, this study can be seen as an encouraging starting point, and the reported models could be improved both by enriching the arsenal of the considered descriptors and by including structure-based approaches (such as docking simulations), which can simulate the recognition between the substrate and the involved enzymes. The predictive models could also be improved by extending and refining the collected metabolic data. Enriched metabolic data should maximize the chemical space covered by the collected reactions, thus extending the predictive power of the developed models. In addition, enriched metabolic data could be utilized to generate suitable external sets for a more precise validation and tuning of the selected predictive models.

## Figures and Tables

**Figure 1 ijms-24-11064-f001:**
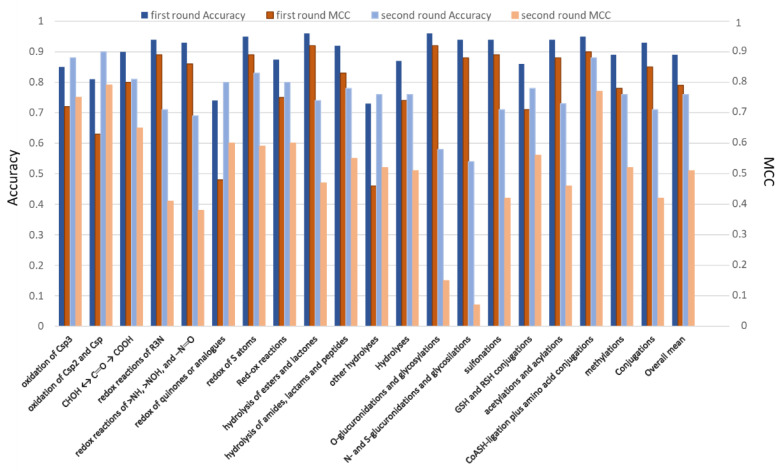
Performances (as described by Accuracy and MCC values) reached by the classification models for the classes of metabolic reactions in the two rounds of predictions. The average values for the three major classes (i.e., redox, hydrolyses and conjugations) as well as the overall means are also reported.

**Figure 2 ijms-24-11064-f002:**
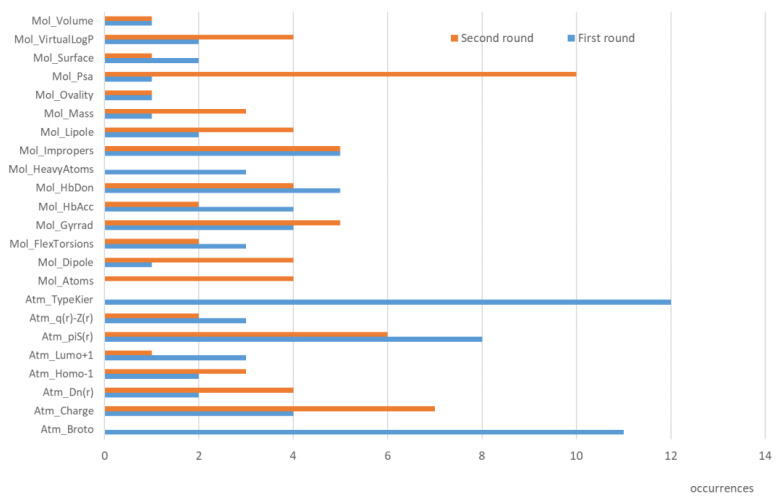
Occurrences of the descriptors as derived by analysis of feature importance for the predictive models for the classes of reactions in the two rounds of predictions.

**Figure 3 ijms-24-11064-f003:**
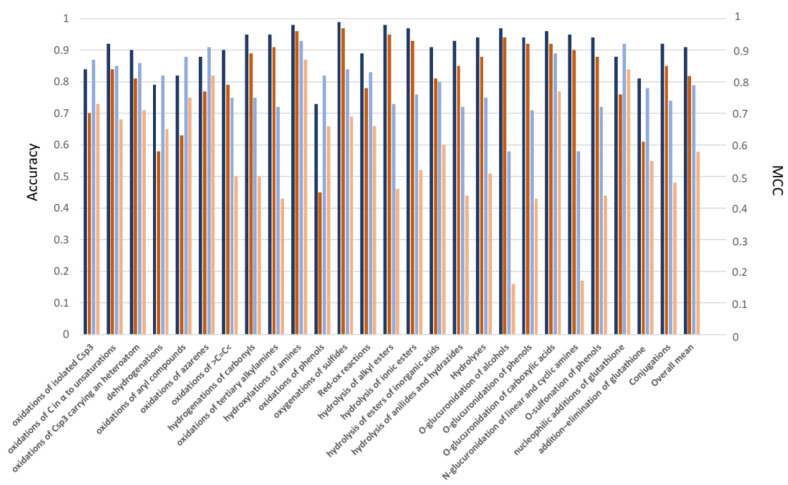
Performances (as described by Accuracy and MCC values) reached by the classification models for the subclasses of metabolic reactions in the two rounds of predictions. The color code of the columns is the same as Figure 1 (i.e., blue and brown for accuracy and MCC of the first round; azure and light brown for accuracy and MCC of the second round). The average values for the three major classes (i.e., redox, hydrolyses and conjugations) as well as the overall means are also reported.

**Figure 4 ijms-24-11064-f004:**
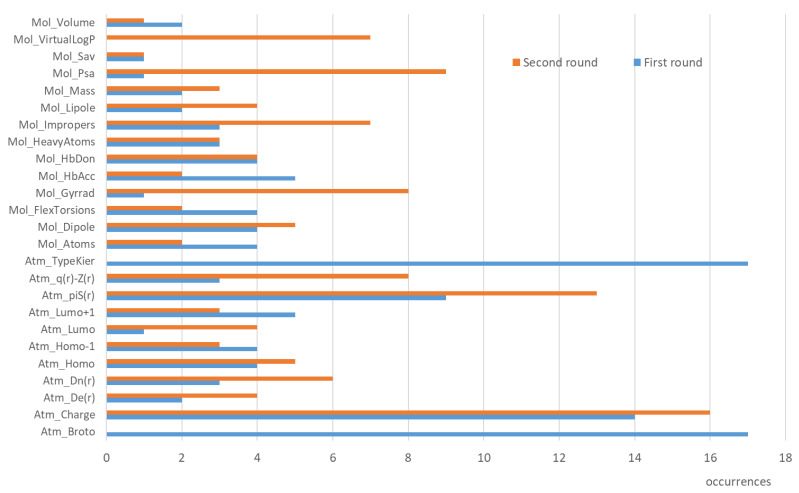
Occurrences of the descriptors as derived by analysis of feature importance for the predictive models for the subclasses of reactions in the two rounds of predictions.

**Figure 5 ijms-24-11064-f005:**
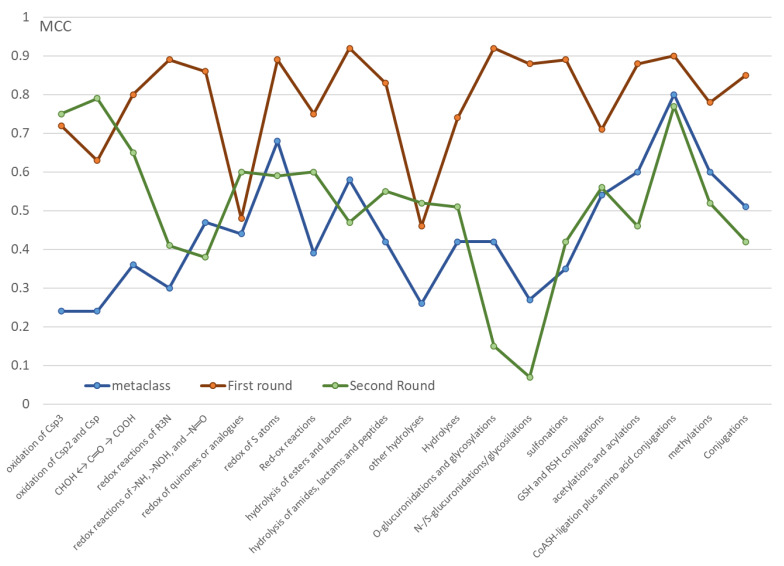
Comparison of the performances (expressed as MCC values) of the models generated by the previous MetaClass project with those here developed by the two rounds of predictions.

## Data Availability

All data are reported in the manuscript and in the Appendix A. The developed predictors are included as scripts.

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
