# Peer review of "MetaSpot: A General Approach for Recognizing the Reactive Atoms Undergoing Metabolic Reactions Based on the MetaQSAR Database"

_ijms, 2023, doi:10.3390/ijms241311064_

Round 1
Reviewer 1 Report
The Authors present their work on the prediction of metabolism for most metabolic reactions using MetaQSAR-based datasets (MetaSpot project). For each considered biotransformation, two types of models were developed and discussed. Most models provided satisfactory performances emphasizing the possibility of conveniently recognizing the reactive atoms. Overall, the manuscript is well-written, clearly structured, and interesting for a readership with a background in drug metabolism and toxicity. The issues specified below should be carefully addressed in a revised version.
- Clearly state the aim of the work in the Introduction section.
- Line 35: Explain ‘ADME/Tox’ at the first use.
- Lines 39-41: I think that reference is needed here.
- Be consistent in your writing, e.g., ‘in-silico’ or ‘in silico’, dot after or before the reference number in the text, ‘red-ox’ or ‘redox’.
- Lines 224 and 225: Typos: ‘woth’ (line 224), ‘confirned’ (line 225), ‘asses’ (line 398).
- Some editing errors appear in the References section.
- If the authors were to be tempted to write in one/two sentences a "take home message" from their work what would it be? Add some future perspectives in the Conclusions section.
English is fine.
Author Response
Dear Editor,
We thank you and the reviewer #1 for the valuable suggestions. Here is a description of the amendments made in the revised version according to the requests of the referee.
Response to reviewer 1 comments
The Authors present their work on the prediction of metabolism for most metabolic reactions using MetaQSAR-based datasets (MetaSpot project). For each considered biotransformation, two types of models were developed and discussed. Most models provided satisfactory performances emphasizing the possibility of conveniently recognizing the reactive atoms. Overall, the manuscript is well-written, clearly structured, and interesting for a readership with a background in drug metabolism and toxicity. The issues specified below should be carefully addressed in a revised version.
- Clearly state the aim of the work in the Introduction section.
The last paragraph of the Introduction was reworded to better clarify the aim of the work.
- Line 35: Explain ‘ADME/Tox’ at the first use.
Done.
- Lines 39-41: I think that reference is needed here.
A reference has been added.
- Be consistent in your writing, e.g., ‘in-silico’ or ‘in silico’, dot after or before the reference number in the text, ‘red-ox’ or ‘redox’.
Corrected.
- Lines 224 and 225: Typos: ‘woth’ (line 224), ‘confirned’ (line 225), ‘asses’ (line 398).
Corrected.
- Some editing errors appear in the References section.
The references were carefully checked.
- If the authors were to be tempted to write in one/two sentences a "take home message" from their work what would it be? Add some future perspectives in the Conclusions section.
A final perspective paragraph was added in the conclusions.
We do believe that the revised version fully answers the referee's comments and we look forward to hearing from you.
Best regards
Alessandro Pedretti

Reviewer 2 Report
In this manuscript, Mazzolari et al. developed a new generalized framework, termed MetaSpot. It can be used to predict drug metabolism using advanced machine learning approaches. Based on the benchmarking provided by the study, and the data trained on the MetaQSAR, their method can reach a good performance. Overall, this is a valuable work and I have two comments:
1, What would be the results if the reaction was not included in the MetaQSAR database? It would be interesting to see the authors’ opinions on this topic.
2, In practice, how many errors can we tolerate for the prediction? If the prediction is not accurate enough, can we obviously tell it, or do we have to validate using some experiments? I look forward to some discussion on this.
Author Response
Dear Editor,
We thank you and the reviewer #2 for the valuable suggestions. Here is a description of the amendments made in the revised version according to the requests of the referee.
Response to reviewer 2 comments
In this manuscript, Mazzolari et al. developed a new generalized framework, termed MetaSpot. It can be used to predict drug metabolism using advanced machine learning approaches. Based on the benchmarking provided by the study, and the data trained on the MetaQSAR, their method can reach a good performance. Overall, this is a valuable work and I have two comments:
1, What would be the results if the reaction was not included in the MetaQSAR database? It would be interesting to see the authors’ opinions on this topic.
Clearly, the models can be trained only on the metabolic reactions included in the training set. The lack of relations between performances and number of instances in each predicted class suggests that all considered training sets cover a significant part of the chemical space of the corresponding substrates. Nevertheless, we cannot exclude that enriched and refined training sets could provide better models and indeed added in the Conclusions the continuous extension of the MetaQSAR database is described among the ideas for future studies.
2, In practice, how many errors can we tolerate for the prediction? If the prediction is not accurate enough, can we obviously tell it, or do we have to validate using some experiments? I look forward to some discussion on this.
As mentioned in the Conclusions, a proper validation of the proposed models should involve the use of external sets of metabolic data. Along with assessing the predictive power, the external validation should allow a more accurate tuning of the models by selecting those offering the best performance in this kind of validation. In the reported study, we decided to validate the models by cross-validation only, a choice which allowed to utilize all the collected metabolic data in the training phase so as to cover the widest possible chemical space of the substrates (see point 1). A sentence detailing this choice was added under Methods. The extension of the MetaQSAR database is also planned to collect the necessary external datasets to be utilized in the following studies.
We do believe that the revised version fully answers the referee's comments and we look forward to hearing from you.
Best regards
Alessandro Pedretti
